# Sevoflurane Induces a Cyclophilin D-Dependent Decrease of Neural Progenitor Cells Migration

**DOI:** 10.3390/ijms24076746

**Published:** 2023-04-04

**Authors:** Pan Lu, Feng Liang, Yuanlin Dong, Zhongcong Xie, Yiying Zhang

**Affiliations:** 1Department of Anesthesia, The Second Affiliated Hospital of Xi’an Jiaotong University, Xi’an 710004, China; 2Department of Anesthesia, Critical Care and Pain Medicine, Massachusetts General Hospital and Harvard Medical School, Charlestown, MA 02129, USA

**Keywords:** sevoflurane, cyclophilin D, neural progenitor cells, migration, doublecortin

## Abstract

Clinical studies have suggested that repeated exposure to anesthesia and surgery at a young age may increase the risk of cognitive impairment. Our previous research has shown that sevoflurane can affect neurogenesis and cognitive function in young animals by altering cyclophilin D (CypD) levels and mitochondrial function. Neural progenitor cells (NPCs) migration is associated with cognitive function in developing brains. However, it is unclear whether sevoflurane can regulate NPCs migration via changes in CypD. To address this question, we treated NPCs harvested from wild-type (WT) and CypD knockout (KO) mice and young WT and CypD KO mice with sevoflurane. We used immunofluorescence staining, wound healing assay, transwell assay, mass spectrometry, and Western blot to assess the effects of sevoflurane on CypD, reactive oxygen species (ROS), doublecortin levels, and NPCs migration. We showed that sevoflurane increased levels of CypD and ROS, decreased levels of doublecortin, and reduced migration of NPCs harvested from WT mice in vitro and in WT young mice. KO of CypD attenuated these effects, suggesting that a sevoflurane-induced decrease in NPCs migration is dependent on CypD. Our findings have established a system for future studies aimed at exploring the impacts of sevoflurane anesthesia on the impairment of NPCs migration.

## 1. Introduction

Approximately six million children in the United States undergo surgery with anesthesia each year [1]. Population studies have suggested that early and repeated, but not single, exposure to anesthesia and surgery may increase the risk of cognitive impairment in children ([2,3,4,5], reviewed in [6,7]). 

Sevoflurane, a commonly used anesthetic in pediatric patients, has been shown to induce neurotoxicity and cognitive impairment in rodents ([8,9,10,11,12,13], reviewed in [6]). Specifically, multiple exposures to sevoflurane have been shown to suppress neurogenesis in hippocampal neural stem/progenitor cells in vitro and in rodent models [14,15,16,17]. However, the underlying mechanism by which sevoflurane influences neurogenesis is not yet fully understood.

Previous studies have demonstrated that anesthesia-induced neurotoxicity and cognitive impairment in both young and adult mice are associated with mitochondrial dysfunction, particularly the opening of the mitochondrial permeability transition pore (mPTP) [18]. In addition, anesthesia has been shown to affect neurogenesis [16,17,19] and synaptogenesis [20,21] in the developing brain. These findings suggest that mitochondrial dysfunction plays a critical role in anesthesia-induced neurotoxicity and cognitive impairment in young mice.

Cyclophilin D (CypD) is a protein located in the mitochondria and is part of the mPTP, and CypD regulates mPTP opening and is associated with mitochondrial dysfunction and cell death [22,23]. Increased CypD expression has been observed in the brain of adult AD transgenic mice, which may contribute to the disease’s progression [24,25,26]. CypD deficiency stabilizes mitochondrial function and may improve cognitive and synaptic function in old AD transgenic mice [24,25,26,27]. 

Sevoflurane anesthesia can cause mitochondrial dysfunction and increase CypD expression in the hippocampus, and CypD deficiency may protect against sevoflurane-induced cognitive impairment [14]. Previous studies have also demonstrated that anesthesia can damage neurogenesis by decreasing the self-renewal capacity of neural stem cells [28,29,30,31]. 

In addition, anesthesia has been shown to inhibit neural progenitor cells (NPCs) self-renewal, as well as neurogenesis and synaptic plasticity [14,32]. Despite these findings, the impact of CypD on the migration of NPCs in the developing brain is unclear. 

Finally, in our previous work, we detected the effect of sevoflurane on the mitochondrial permeability transit pore (mPTP) in NPCs and demonstrated that sevoflurane reduced the binding of CypD with adenine nucleotide translocase (ANT, the other component of mPTP) and caused the mPTP opening [14].

To gain a comprehensive understanding of how anesthesia affects changes in mitochondrial proteins and to investigate the potential link between mitochondrial function and NPCs migration, we also used mass spectrometry (MS)-based proteomics to identify changes in protein abundance, which can provide information on both metabolic and cellular pathways as well as predicted biochemical responses. Specifically, we aimed to identify potential changes in mitochondrial proteins and determine whether sevoflurane-induced elevation of CypD could lead to decreased expression of doublecortin (DCX), the marker of cell migration [33], and reduced migration of NPCs. 

Previous studies have shown that mitochondrial reactive oxygen species (ROS) generation is dependent on CypD expression [34]. Thus, in the present study, we determined the effects of sevoflurane on the levels of both ROS and CypD in NPCs.

## 2. Results

We performed the in vitro studies using NPCs in the present study. To confirm that the cells we used in subsequent studies were indeed NPCs, we first observed that these cells could grow into neurospheres. Further analysis revealed that most of these neurospheres stained positive for nestin, a common marker of NPCs, as well as SOX2, [SRY (sex determining region Y)-box 2]. We also conducted immunofluorescence staining and found that the neurospheres were able to differentiate into neurons (as marked by the neuronal marker β-Tubulin) and astrocytes (as marked by Glial fibrillary acidic protein, GFAP). These results are depicted in Figure 1.

### 2.1. Sevoflurane Induced the Decrease of Migration in NPCs Harvested from WT Mice

We then exposed the NPCs harvested from wild-type (WT) mice to 4.1% sevoflurane for 4 h, and the treated NPCs were subjected to wound healing assay. We found that the sevoflurane treatment led to a decrease in the distance the cells were able to move at 24 h after the sevoflurane treatment (Figure 2A). Quantification of the migration distance showed that the sevoflurane treatment significantly reduced cell migration at 24 h after the sevoflurane treatment (Figure 2B). In addition, we conducted a transwell assay and observed a decrease in the number of cells that were able to migrate after 24 h of sevoflurane treatment (Figure 2C,D).

### 2.2. Sevoflurane Increased CypD and ROS Levels and Decreased DCX Levels in NPCs Harvested from WT Mice

We first found that sevoflurane treatment increased ROS levels compared to the control condition in NPCs harvested from WT mice (Figure 3A). We next analyzed the levels of CypD protein in the NPCs based on the immunohistochemistry staining and Western blot. To examine CypD levels, we first stained mitochondria using Complex V as a marker and then analyzed CypD levels in the mitochondria of NPCs harvested from WT mice. Immunofluorescence staining of CypD revealed a significant increase in CypD levels in the mitochondria of NPCs harvested from WT mice following the sevoflurane treatment compared to the control condition (Figure 3B). Immunoblotting of CypD showed that sevoflurane treatment resulted in increased CypD levels compared to the control condition (Figure 3C). There was no significant difference in β-Actin levels between these two groups. Quantification of the Western blot demonstrated a significant increase in CypD levels in NPCs harvested from WT mice following the sevoflurane treatment (Figure 3D, * *p* = 0.029). Conversely, sevoflurane treatment led to a decrease in DCX levels compared to the control condition (Figure 3E). Quantification of the Western blot demonstrated a significant decrease in DCX levels in NPCs harvested from WT mice following the sevoflurane treatment (Figure 3F, ** *p* = 0.004).

### 2.3. Sevoflurane Induced a CypD-Dependent Impairment of Migration of NPCs

To investigate the role of CypD in sevoflurane-induced effects on NPCs, we used NPCs harvested from CypD knockout (KO) mice in the present study. We found that there were no visible changes in ROS levels (Figure 4A) or DCX levels (Figure 4B,C) in NPCs harvested from CypD KO mice between the sevoflurane treatment and control condition. Additionally, we conducted a transwell assay and observed no visible change in the number of cells that were able to migrate after 24 h of sevoflurane treatment in NPCs harvested from CypD KO mice compared to the control condition (Figure 4D,E).

### 2.4. Sevoflurane Induced a CypD-Dependent Decrease of DCX-Positive Cells in Hippocampus of Young WT Mice

Given the in vitro findings that sevoflurane induced a CypD-dependent effect in impairing the migration of NPCs, next, we set out to determine the in vivo relevance of these findings by employing young WT and CypD KO mice.

To have a comprehensive understanding of the protein changes associated with sevoflurane anesthesia, we first determined the profiles of mitochondrial proteomes in the hippocampus of young WT mice following sevoflurane anesthesia, utilizing the TMT 10-plex tagging approach (Figure 5A). We performed the mitochondrial proteome analysis with a false discovery rate (FDR) < 1% for both peptide and protein identification (Appendix A), and we chose the proteins which had more than a two-fold increase after the sevoflurane anesthesia. We identified 12 proteins that were associated with peptidyl-prolyl cis-trans isomerase activity (Figure 5B and Table 1). Among them, the peptidyl-prolyl cis-trans isomerase F (*Ppif*, CypD) had a 3.9-fold upregulation in the hippocampus of the young WT mice following the sevoflurane anesthesia compared to the control condition.

Our previous studies have demonstrated that the opening of the mitochondrial permeability transition pore (mPTP) is involved in the anesthesia-induced mitochondrial dysfunction [18]. CypD, a mitochondrial matrix protein, plays an essential role in the regulation of the opening of mPTP and has been identified as a potential target of anesthesia-induced neurotoxicity in the developing brain [14]. Given the finding that sevoflurane anesthesia induced a 3.9-fold upregulation of CypD levels in the hippocampus of young WT mice, we measured the CypD protein levels by Western blot in the hippocampus of young WT mice. Immunoblotting of CypD revealed that the sevoflurane anesthesia (2% and 2 h daily for 3 days on postnatal 6, 7, and 8) led to increased visibility of the bands representing CypD as compared to the control condition at postnatal 8 (Figure 5C). There were no significant differences in β-actin levels in the hippocampus between the sevoflurane-treated mice and the mice in the control group.

Furthermore, we analyzed DCX-positive cells in the hippocampus of postnatal 8 WT mice after the sevoflurane anesthesia. Immunofluorescence staining of DCX demonstrated that there were significantly fewer DCX-positive cells in the hippocampus of the postnatal 8 WT mice after the sevoflurane anesthesia compared to the control condition (Figure 6A). Quantification of the images revealed that there were significant decreases in the number of DCX-positive cells in the hippocampus of sevoflurane-treated WT mice compared to the control condition (Figure 6B, * *p* = 0.043).

Next, we assessed the effects of sevoflurane anesthesia on the number of DCX-positive cells in the hippocampus of young CypD KO mice. In contrast to young WT mice, the same sevoflurane anesthesia did not significantly change the number of DXC-positive cells in the hippocampus of the young CypD KO mice on postnatal day 8 (Figure 6C, D). These data demonstrated that the KO of CypD might attenuate the sevoflurane-induced impairment of migration of NPCs. These in vivo results in both WT and CypD KO young mice were consistent with the in vitro findings in the NPCs harvested from WT mice and NPCs harvested from CypD KO mice.

## 3. Discussion

Overall, the study highlighted the potential adverse effects of sevoflurane anesthesia on the migration of NPCs and mitochondrial function, which may have implications for anesthesia neurotoxicity in the developing brain. Future studies will further test the hypothesis generated in this proof-of-concept study. Specifically, sevoflurane led to increases in ROS and CypD levels in vitro in NPCs and in vivo in the hippocampus of young mice. This upregulation of CypD then contributes to the sevoflurane-induced decrease of migration of NPCs in vitro and the number of DCX-positive cells in the hippocampus of young mice in vivo. These results suggest that sevoflurane can induce a CypD-dependent and DCX-related impairment of NPCs migration.

In the in vitro NPCs studies, we demonstrated that the sevoflurane treatment could decrease the migration of NPCs harvested from WT mice (Figure 2) as well as increase levels of ROS and CypD and decrease DCX levels in the NPCs harvested from WT mice (Figure 3). More importantly, the same sevoflurane treatment did not significantly change the levels of ROS and DCX nor the migration of NPCs harvested from CypD KO mice (Figure 4). Together, these results suggest that sevoflurane can induce a CypD-dependent impairment of the migration of NPCs.

Then, in the in vivo relevance study in both WT and CypD KO mice (Figure 5 and Figure 6), the sevoflurane anesthesia was found to specifically increase the protein levels of CypD in the hippocampus of young mice and NPCs. The study also showed that sevoflurane anesthesia decreased the number of DCX-positive cells in the hippocampus of WT mice. DCX is the marker of migration of cells [18]; thus, these results suggest that sevoflurane can also impair the migration of NPCs in the hippocampus of mice.

Moreover, sevoflurane anesthesia did not significantly change the DCX-positive cells in the hippocampus of CypD KO mice. Taken together, these results suggest that sevoflurane anesthesia can induce a CypD-dependent and DCX-related impairment of NPCs migration in the hippocampus of young mice.

Our previous studies have shown that sevoflurane anesthesia might induce a CypD-dependent impairment of neurogenesis and cognitive impairment in the developing brain [14]. The current study is the first to suggest that CypD also contributes to the sevoflurane-induced decrease in the migration of NPCs. Further studies are needed to confirm these findings and to explore potential strategies for mitigating the adverse effects of sevoflurane anesthesia on neurotoxicity and mitochondrial dysfunction in the developing brain.

Using proteomics, the study found that sevoflurane may affect peptidyl-prolyl cis-trans isomerase proteins, especially cyclophilin D (CypD) (Figure 5), as well as other proteins associated with sevoflurane (Figure 5, Table 1, and Appendix A). Although we only focused on the CypD experiment in the present study, the findings from the proteomics will help us to design additional studies to further investigate the underlying mechanism of anesthesia neurotoxicity in the developing brain.

CypD is a mitochondrial matrix protein that plays a critical role in regulating mitochondrial function and has been implicated in several neurological diseases [23,24,25,26]. CypD deletion prevents cognitive impairment and mitochondrial dysfunction associated with aging by preventing mPTP opening [27]. CypD also protects brain mitochondria against the permeability transition, which contributes to neuronal and oligodendrocyte injury in various neurodegenerative diseases [23,24,25,26]. However, the role of CypD in hypoxic-ischemic brain injury shifts from promoting to protecting against injury during development [35]. These findings suggest that CypD plays a complex role in neuroprotection and neurodegeneration, with its effects depending on the context and disease stage. Zhang et al. demonstrated that CypD contributes to anesthesia neurotoxicity in the developing brain by inducing mitochondrial dysfunction and decreasing neurogenesis in NPCs [14]. The findings from the current studies showed, for the first time, that sevoflurane may also induce a CypD-dependent and DCX-related decrease in the migration of NPCs (Figure 7).

It is not known at present how CypD affects DCX levels. We hypothesize that the increases in CypD levels can cause mitochondrial dysfunction, which attenuates the migration of NPCs, marked by decreases in DCX levels. Future studies will test this hypothesis both in NPCs in vitro and in young mice in vivo.

NPCs migration is the prerequisite for the development of the embryonic nervous system. In both embryonic and adult nervous systems, most NPCs need to migrate a certain distance to reach their functional sites [36]. Defective NPCs migration is closely associated with multiple neurodegenerative diseases and cognitive impairments [37,38].

In our previous study, we demonstrated that sevoflurane anesthesia increased CypD levels and reduced NPCs proliferation in WT young mice or NPCs harvested from WT mice but not in CypD KO mice or NPCs harvested from CypD KO mice. These data suggest that the sevoflurane might induce a CypD-dependent inhibition of cell proliferation [14]. In the present study, we further found that sevoflurane might also induce a CypD-dependent inhibition of the migration of NPCs.

The studies have several limitations. Firstly, we only investigated the effects of sevoflurane in the present study, but not the other anesthetics such as isoflurane and desflurane. Our future studies will include the comparison among sevoflurane, isoflurane, and desflurane on NPCs migration. Secondly, we did not assess the potential sex-dependent effects of anesthesia neurotoxicity in developing brains in the current studies, as it is challenging to determine the sex of 6-day-old mice accurately. We plan to explore this aspect in future studies by utilizing established systems to determine sex-dependent effects. Finally, we did not determine the cognitive function in young mice in the present study; however, our previous studies have shown well that the same sevoflurane anesthesia can induce cognitive impairment in young mice [14,39,40,41]. Nevertheless, future studies still need to directly link the sevoflurane-induced inhibition of NPCs migration and the sevoflurane-induced cognitive impairment in young mice.

In conclusion, our studies suggest that anesthetic sevoflurane can inhibit the migration of NPCs, and such effects are dependent on CypD. These results could promote further research into mitochondria, migration of NPCs, and anesthesia-induced neurotoxicity in the young brain, which could ultimately lead to better postoperative outcomes in children.

## 4. Materials and Methods

### 4.1. Neural Progenitor Cells (NPCs) Culture and Treatment

The protocol use of animals in research and teaching was approved by the Massachusetts General Hospital Standing Committee in Boston. For the experiments, C57BL/6J mice [wild-type (WT)] and CypD homozygous null mice, B6;129-Ppiftm1Maf/J (*Ppif*−/−) mice were used, all of which were obtained from The Jackson Laboratory in Bar Harbor, ME. Primary cultured NPCs were obtained from mice fetuses at embryonic day 18–19 (E18-E19) from timed-pregnant mice under sterile conditions. The mice were bred in-house and kept in a temperature- and humidity-controlled environment (20–22 °C; 12-h light: dark on a reversed light cycle) with free access to food and water. The NPCs were harvested using the methods described in previous studies [14], and 1.0 × 10^6^ cells were seeded in each well of a six-well plate containing 2.5 mL of “complete” proliferation media. The “complete” proliferation media consisted of 50 mL of NeuroCult™ neural stem cell proliferation supplements and 200 μg of human epidermal growth factor (hEGF), which were added to 450 mL of NeuroCult™ neural stem cell basal media to achieve a final concentration of 20 ng/mL of hEGF, 100 units/mL of penicillin, and 100 μg/mL of streptomycin in the culture media. For the in vitro studies, the NPCs were treated with 4.1% sevoflurane plus 21% O_2_ and 5% CO_2_ for 4 h, as described in previous studies [14].

### 4.2. Immunofluorescence Staining

The cultured cells were seeded onto six-well plates, and their differentiation potency was characterized and tested using immunofluorescence staining analysis. To fix the cells, 4% paraformaldehyde (PFA) was used. Mouse monoclonal anti-nestin antibody (1:500, Abcam, Boston, MA, USA), rabbit polyclonal anti-glial fibrillary acidic protein (GFAP) antibody (1:500, Abcam), and mouse monoclonal anti-β-tubulin III antibody (1:500, Abcam) were used for NPCs identification. D4′, 6′-diamidion-2′-phenylindole (DAPI, 1:1000, Abcam) was used to counterstain the nuclei. To detect CypD and Complex V (mitochondrial marker), anti-CypD (1:200, ab110324, Abcam) and anti-complex V (1:200, 45, 9000, Invitrogen, Carlsbad, CA, USA) antibodies were used. After fixing the cells for 20 min, they were placed in 0.1% Triton X-100 for 30 min, followed by a blocking buffer (10% goat serum) for 1 h. The cells were then incubated with primary antibodies overnight at 4 °C. Samples were incubated in secondary antibodies [fluorescein isothiocyanate (FITC) or tetramethylrhodamine isothiocyanate (TRITC)-conjugated IgG] for 2 h, followed by DAPI incubation for 5 min at room temperature. The slides were rinsed with DPBS, and the images were captured using a laser confocal microscope (TSC SP2, Leica, Manheim, Germany).

### 4.3. Wound Healing Assay

To assess the directional migration of NPCs, a wound healing assay was performed. The NPCs were seeded on poly-D-lysine (PDL) precoated six-well plates at a cell density of 1.0 × 10^6 ^cells/mL in a “complete” proliferation medium and allowed attaching for 12 h. After cell attachment, plates with consistent cell density were selected for the next experiments. After cell attachment, a scratch was created on the monolayer using a sterile plastic 200 μL micropipette tip in each cultured well. The cells were washed twice in a warm serum-free medium to remove cellular debris and floating cells from scratches and then exposed to 4.1% sevoflurane plus 21% O_2_ and 5% CO_2_ for 4 h. The scratch wounds were visualized with a phase contrast light microscope. Photographs were captured at two time points (0 h and 24 h after sevoflurane exposure). The NPCs were kept in the incubator between the photographs within the “complete” proliferation medium. The measurement was conducted by using Image J software (Version 1.8.0). The distance between the two separated sides at 0 h was used as a reference, and subsequent changes in the distance over time were measured as the cell migration distance. Result was quantified by determining the percent scratch closed [(initial scratch width- final scratch width)/initial scratch width × 100%].

### 4.4. Transwell Migration Assay

To evaluate NPCs migration, an 8-mm pore-size transwell system (Costar, Cambridge, MA, USA) coated with poly-D-lysine (PDL, Sigma, St. Louis, MO, USA) at 100 μg/mL in PBS overnight was used. NPCs were dissociated into single cells and resuspended in a “complete” proliferation medium at a density of 1.0 × 10^6^ cells/mL. Then, 200 μL of cell suspension was loaded into the top chamber of the transwell, and 600 μL of “complete” proliferation medium containing 10% FBS was loaded into the bottom chamber. The cells were cultured for 12 h to form an adherent monolayer culture. After sevoflurane treatment, the cells were cultured for another 24 h; then, the cells on top of the membrane were removed by a cotton swab, and the migrated cells on the bottom of the membrane were fixed with 4% PFA and stained with either DAPI or 0.1% crystal violet. The migrated cells were photographed under a microscope, and for each insert, 10 fields were randomly captured, and the cell number was counted using Image-Pro Plus 6.0.

### 4.5. Western Blot Analysis

Western blot analysis was performed as described [42]. Specifically, anti-CypD (1:1000, ab110324, Abcam) and anti-DCX (1:500, ab113435, Abcam) were used to detect CypD and DCX separately. β-actin antibody (1:5000, 42 kDa, Sigma, St. Louis, MO, USA) was used to detect β-actin as a loading control. Quantification of Western blots was performed as described previously [42]. A protein levels of 100% indicates control levels. Signal intensity was analyzed using a Bio-Rad (Hercules, CA, USA) image program. The Western blots were quantified using two steps. First, β-actin levels were used to normalize (e.g., determining the ratio of CypD levels to β-actin levels) for any loading differences in total protein amounts. Second, changes in CypD levels in treated cells were presented as percentages of the corresponding levels in the control cells. Each band in the Western blot represented an independent experiment. The results were averaged by six independent experiments.

### 4.6. Reactive Oxygen Species (ROS) Measurement

ROS measurement was performed as described previously [18]. Briefly, prepare NPCs cells by culturing them in a black 96-well cell culture plate overnight in the incubator. Prepare the DCFH-DA/media solution by dissolving DCFH-DA in the cell culture media to a final concentration of 10 μM. Protect the solution from light during the preparation and use. Remove the cell culture media from the NPCs cells and add the DCFH-DA/media solution to each well. Incubate the cells at 37 °C for 30 min to allow the cells to uptake the DCFH-DA. After 30 min, remove the DCFH-DA/media solution and wash the cells with fresh media to remove any excess DCFH-DA. The DCFH-DA loaded NPCs cells were then exposed to 4.1% sevoflurane for 4h. The treated cells were lysed by adding 100 μL of cell lysis buffer to each well. Mix the lysate thoroughly and incubate for 5 min at room temperature away from light. Finally, the fluorescence was read with a fluorometric plate reader at 480 nm/530 nm within 10–15 min. The amount of fluorescence is proportional to the amount of ROS present in the cells.

### 4.7. Mice Anesthesia

The protocol of the animal study was approved by the Massachusetts General Hospital Standing Committee on Animals (Boston, MA, USA) on the use of Animals in Research and Teaching.

We made every effort to minimize the number of mice used and obtained both WT and CypD KO mice from our breeding. The mice were housed in a controlled environment with free access to water and food, and we maintained the temperature at 20–22 °C on a reversed light cycle of 12 h light and dark. Due to the difficulty of determining the sex of young mice, we did not allocate an equal number of female and male mice to the anesthesia or control group and did not assess the sex-dependent effects of anesthesia neurotoxicity in the present study. The mice were randomly assigned to either the anesthesia group or the control group. The mice in the anesthesia group received 3% sevoflurane plus 60% oxygen (balanced with nitrogen) for two hours on postnatal day(P) P6, P7, and P8, as described in our previous studies [39,40,41,43].

### 4.8. Brain Tissue Collection

Mice were sacrificed by carbon dioxide and decapitation at the end of the experiments. We used half of the harvested brain tissues for the immunohistochemistry staining and half of them for the Western blots, according to our previous studies [14]. We used different mice and harvested their hippocampus tissues to obtain the mitochondria for the mass spectrometry studies.

### 4.9. A Tandem Mass Tag (TMT)-Labeling and Sample Cleaning up for MASS Spectrometry Study

In the current study, we utilized TMT for the relative quantitation of proteins present in multiple samples. Isobaric stable isotope tags were used to label the peptides, which fragmented into reporter ions upon collision-induced dissociation, allowing for quantitation. Mitochondria samples were obtained from the hippocampus tissues of young mice, and 10-plex TMT tags were employed. The labeling of tryptic peptides was carried out according to the manufacturer’s instructions (Thermo Fisher Scientific, Waltham, MA, USA). Briefly, TMT reagents (0.8 mg) were dissolved in 41 μL of anhydrous acetonitrile, and aliquots of samples were incubated with TMT reagents for one hour at room temperature. The reactions were quenched by adding 8 μL of 5% hydroxylamine solution and reacted for 15 min. The combined TMT-labeled samples were dried under Speed Vac, reconstituted with trifluoroacetic acid solution, and desalted using an Oasis HLB 96-well Elution plate (Waters Corporation, Milford, MA, USA) before LC-MS/MS analysis.

### 4.10. Proteomic Data Analysis

The data were searched twice against a concatenated decoy Swiss Prot mouse database v 3.46 using BioWorks v 3.3.1 (Thermo Fisher Scientific) and SEQUEST v 28 (Thermo Fisher Scientific). The parameters for the SEQUEST search were the same as described previously [44]. Peptides were filtered and assembled into proteins using DTASelect v 1.9. Non-mitochondrial protein contaminants with altered protein levels and inaccurately quantified proteins were not considered in the analysis. Significant changes in protein levels were determined based on a fold change > 2.0 (upregulated) and a *p*-value < 0.05 in the experiment.

### 4.11. Immunohistochemistry Staining

To assess the effects of sevoflurane on DCX levels in mouse brain tissue, immunohistochemistry was performed [44]. Specifically, on P8, mice were decapitated, and their brain tissues were harvested and fixed in formalin for 30 min at room temperature. Brain sections were stained with anti-DCX antibody (1:200, ab113435, Abcam) and examined under a 10× and 40× objective using a KEYENCE BZ-9000E all-in-one fluorescence microscope. Regions of interest in the dentate gyrus, including the outer and inner halves of the granular layer, were analyzed using the software provided by KEYENCE Corporation of America. The software enabled us to quantify red colors that corresponded to DCX staining or DAPI staining. We then counted the numbers of DCX-positive cells and DCX/DAPI-double-positive cells in each region per unit area using Image-J (NIH Image 1.62, Bethesda, MD, USA). The mean number of DCX/DAPI double-positive cells was calculated from five to six brain sections for each animal, with N = 3 mice for each group.

### 4.12. Statistical Analysis

Based on our previous studies, sufficient power to detect a significant effect should be achieved using the wound healing assay, transwell assay, Western blot, and fluorescence staining studies, and we used 6 samples per group for the former three assays and 3 samples per group for the latter. We used the student’s *t*-test with Bonferroni post-hoc adjustment to compare the difference between the two groups. Data are presented as mean ± standard deviation (SD). We defined statistical significance as *p* < 0.05 and *p* < 0.01, and significance testing was two-tailed. For Bonferroni corrections, the adjusted *p*-values, calculated by dividing real *p*-values by sample size, are reported. Statistical analysis was conducted using GraphPad Prism software (version 8.0) and SPSS statistical software (version 21.0).

## Figures and Tables

**Figure 1 ijms-24-06746-f001:**
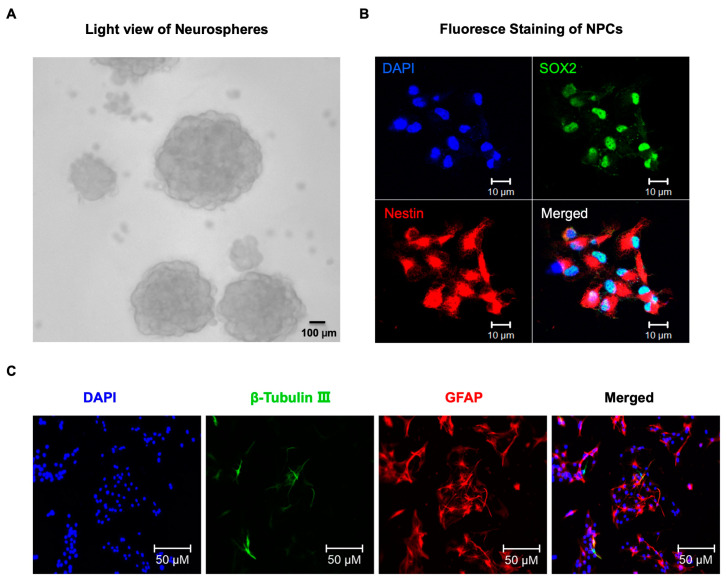
Identification of neural progenitor cells (NPCs). (**A**) Light view of NPCs, Scale bar = 100 μm. (**B**) Immunostaining of NPCs for DNA marker DAPI (blue), the NPCs markers SOX2 (green), and nestin (red). Scale bar = 10 μm. (**C**) Representative immunofluorescence images of NPCs that differentiate into neurons and astrocytes after 3 day’s differentiation culture. Double immunofluorescent staining of cells for β-tubulin III (neuronal marker, green) and GFAP (astrocyte marker, red). The nuclei are counterstained with DAPI (blue). Scale bars = 50 μm.

**Figure 2 ijms-24-06746-f002:**
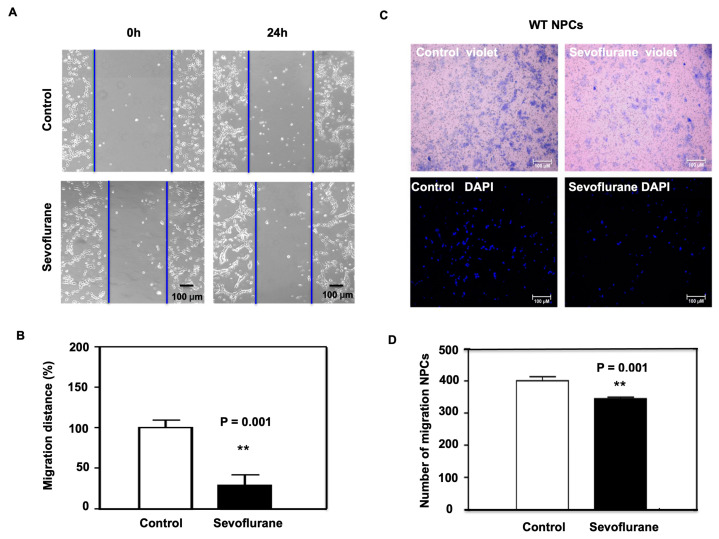
The effects of the sevoflurane treatment on cell migration using wound healing and transwell assays. (**A**) The bright-field images of the wound healing assay demonstrate cell movement 24 h after the sevoflurane treatment compared to the control condition. The blue line indicates the migration distance relative to the control condition. NPCs were maintained in the incubator between inspections. (**B**) Quantification of the migration shows that the sevoflurane treatment significantly reduces the migration distance in the wound healing assay compared to the control condition at 24 h after the sevoflurane treatment (** *p* = 0.001). (**C**) Transwell assay images of migrated cell stained by crystal violet (top) or DAPI (below) 24 h after the sevoflurane treatment. (**D**) Quantification of the transwell assay shows that the sevoflurane treatment significantly decreases the number of migrated cells as compared to the control condition (** *p* = 0.001; N = 10 in each group). Student’s *t*-test was used to analyze the data.

**Figure 3 ijms-24-06746-f003:**
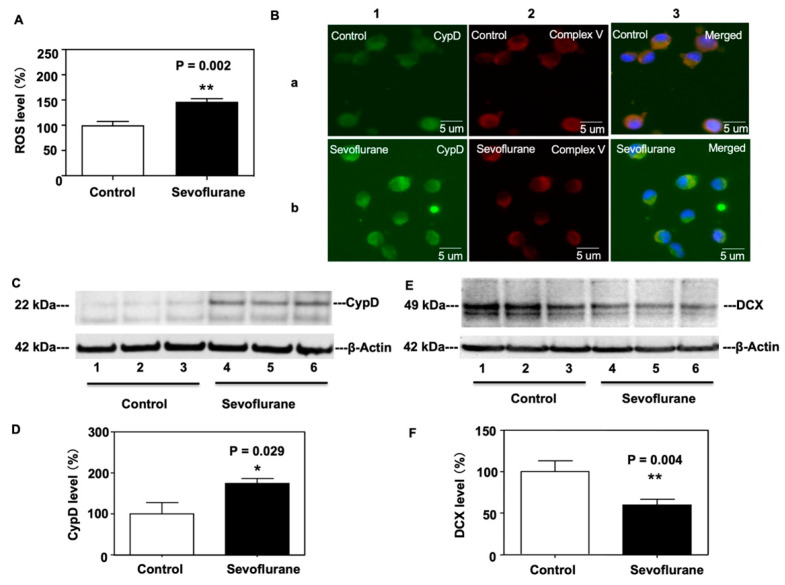
Sevoflurane increases ROS and CypD levels but decreases doublecortin (DCX) levels in NPCs harvested from WT mice. (**A**) Sevoflurane treatment (black bar) increases the levels of ROS compared to the control condition (white bar). (**B**) Representative images of immunofluorescent staining of CypD in NPCs harvested from WT mice. Column 1 is the CypD (green) staining, column 2 is the mitochondrial markers Complex V (red) staining, and column 3 is a merged staining image. Sevoflurane (row b) increases levels of CypD compared to the control condition (row a) in NPCs harvested from WT mice. (**C**) Western blot shows that the sevoflurane treatment (lanes 4 to 6) discernibly increases the levels of CypD compared to the control condition (lanes 1 to 3). There is no significant difference in β-actin levels between the sevoflurane treatment and control condition. (**D**) Quantitation of the Western blot illustrates increased amounts of CypD levels following the sevoflurane treatment compared to the control condition (* *p* = 0.029). (**E**) Sevoflurane treatment (lanes 4 to 6) discernibly decreases doublecortin (DCX) levels compared to the control condition. There is no significant difference in β-actin levels between the sevoflurane treatment and control condition. (**F**) Quantitation of the Western blot illustrates the decreased amounts of DCX levels following the sevoflurane treatment as compared to the control condition (** *p* = 0.004). N = 6–8 in each group. Student’s *t*-test was used to analyze the data.

**Figure 4 ijms-24-06746-f004:**
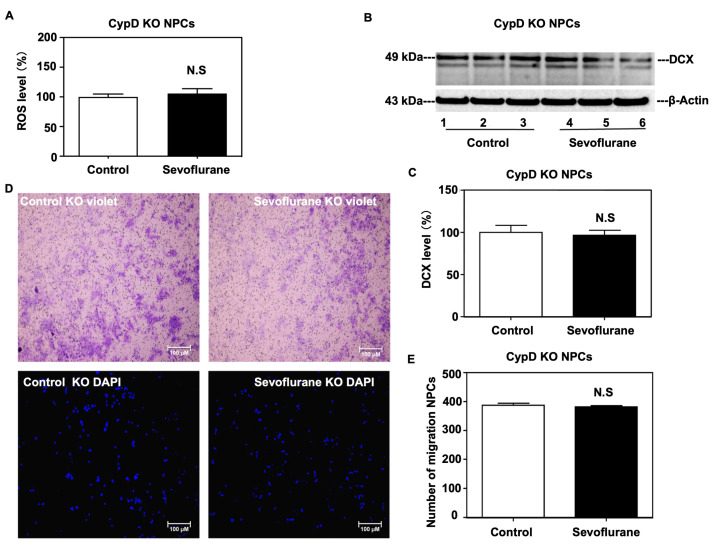
CypD KO mitigates the sevoflurane-induced increases of ROS, decreases of DCX, and inhibition of cell migration in NPCs. (**A**) The sevoflurane treatment does not increase ROS levels compared to the control condition in NPCs harvested from CypD KO mice. Quantitative Western blot (**B**,**C**) analysis shows that the sevoflurane treatment does not decrease DCX levels compared to the control condition in the NPCs harvested from CypD KO mice. There is no significant difference in β-actin levels between each condition. (**D**) The representative transwell assay images of migrated cell stained by crystal violet (top) or DAPI (below) at 24 h after the sevoflurane treatment. (**E**) Quantification of the transwell assay shows that the sevoflurane treatment does not significantly affect the number of migrated NPCs harvested from CypD KO mice compared to the control condition. N = 6–8 in each group. Student’s *t*-test was used to analyze the data. N.S: non-significant differences.

**Figure 5 ijms-24-06746-f005:**
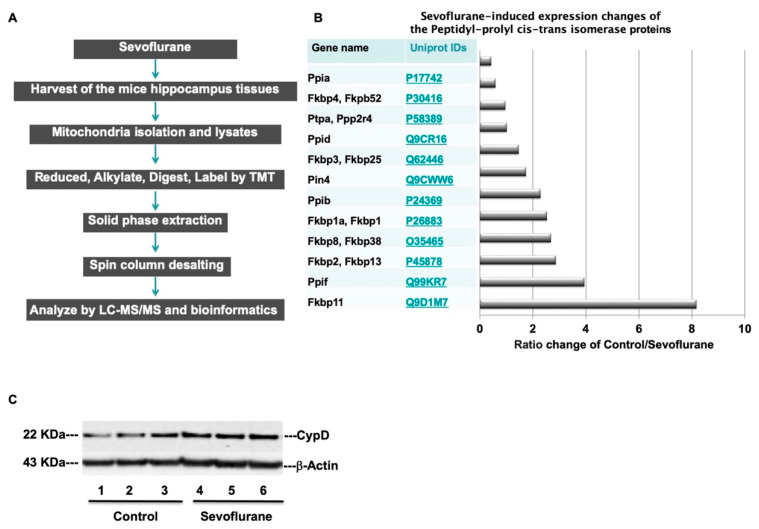
Proteomics analyses identify mitochondrial proteins in the hippocampus of mice. (**A**) Experimental design and workflow for quantitative profiling of mice hippocampus mitochondrial proteomics after the sevoflurane anesthesia. (**B**) Quantitative proteomic analysis of the abundance ratio of Peptidyl-prolyl cis-trans isomerase (*Ppif*) associated proteins after the sevoflurane anesthesia. (**C**) Sevoflurane treatment (lanes 4 to 6) discernibly increases the levels of CypD as compared to the control condition (lanes 1 to 3) in the hippocampus of WT mice; there is no significant difference in β-actin levels between the groups.

**Figure 6 ijms-24-06746-f006:**
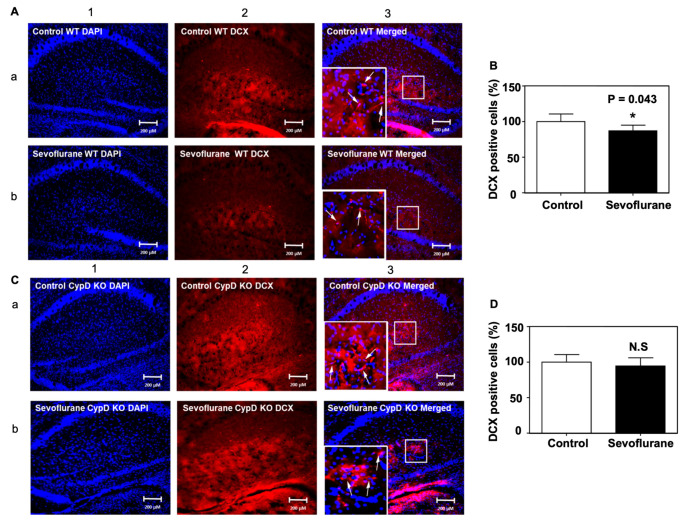
CypD KO mitigates the sevoflurane-induced decrease of DCX-positive cells in the hippocampus of mice. (**A**) Representative images of immunofluorescent staining DCX in the hippocampus of WT mice. Column 1 is the DAPI (blue) nuclear staining, column 2 is the DCX (red) staining, and column 3 is the merged staining image. Sevoflurane (row b) decreases the number of DCX-positive cells compared to the control condition (row a) in the hippocampus of WT mice. (**B**) Quantification of the image shows that sevoflurane (black bar) significantly decreases the number of DCX-positive cells compared to the control condition (white bar) (* *p* = 0.043). (**C**) Representative images of immunofluorescent staining DCX in the hippocampus of CypD KO mice. (**D**) Quantification of the image shows that the sevoflurane anesthesia (black bar) does not change the number of DCX-positive cells in the hippocampus of CypD KO young mice compared to the control condition (white bar). N = 5–6 in each group. Student’s *t*-test was used to analyze the data. The arrows indicate the DCX positive cells in hippocampus. The Box indicates the zoom in image area in staining. N.S: non-significant differences.

**Figure 7 ijms-24-06746-f007:**
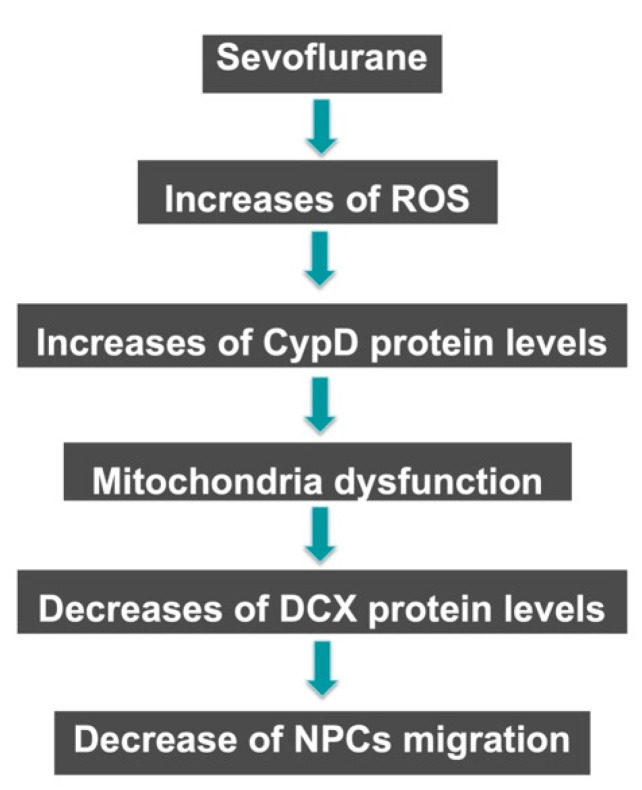
Hypothetical pathway by which sevoflurane decreases mitigation. Sevoflurane triggers the production of reactive oxygen species (ROS) and increases CypD levels to induce mitochondrial dysfunction. Mitochondrial dysfunction leads to a decrease in the levels of DCX, a protein involved in neuronal migration and differentiation, resulting in a decrease in NPCs migration.

**Table 1 ijms-24-06746-t001:** The summary of identified Peptidyl-prolyl cis-trans isomerase proteins regulated by sevoflurane anesthesia.

Uniport IDs	Protein Full Name	Gene Name	MW	Control vs. Sevoflurane Abundance Ratio
Q9D1M7	Peptidyl-prolyl cis-trans isomerase FKBP11	Fkbp11	22.2	8.167487685
Q99KR7	Peptidyl-prolyl cis-trans isomerase F, mitochondrial	Ppif	22	3.931292942
P45878	Peptidyl-prolyl cis-trans isomerase FKBP2	Fkbp2, Fkbp13	15.6	2.857343838
O35465	Peptidyl-prolyl cis-trans isomerase FKBP8	Fkbp8, Fkbp38	44.5	2.669543147
P26883	Peptidyl-prolyl cis-trans isomerase FKBP1A	Fkbp1a, Fkbp1	11.9	2.519260401
P24369	Peptidyl-prolyl cis-trans isomerase B	Ppib	23.7	2.282234957
Q9CWW6	Peptidyl-prolyl cis-trans isomerase NIMA-interacting 4	Pin4	13.8	1.732792463
Q62446	Peptidyl-prolyl cis-trans isomerase FKBP3	Fkbp3, Fkbp25	25.2	1.453658537
Q9CR16	Peptidyl-prolyl cis-trans isomerase D	Ppid	40.7	1.001914608
P58389	Serine/threonine-protein phosphatase 2A activator	Ptpa, Ppp2r4	40.6	0.948455804
P30416	Peptidyl-prolyl cis-trans isomerase FKBP4	Fkbp4, Fkpb52	51.8	0.578207406
P17742	Peptidyl-prolyl cis-trans isomerase A	Ppia	18	0.414748708

The data in Table 1 illustrate the full name, gene name, and other information of the protein identified by using mass spectrometry in the hippocampus tissues obtained from the young mice following the sevoflurane anesthesia.

## Data Availability

Not applicable.

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
