# Peer review of "Sevoflurane Induces a Cyclophilin D-Dependent Decrease of Neural Progenitor Cells Migration"

_ijms, 2023, doi:10.3390/ijms24076746_

Round 1

Reviewer 1 Report

In this article, the authors found that sevoflurane inhibited NPC migration via CypD, which is interesting, but this paper also suffers from the following shortcomings.

1.    In the article, the authors found that Sevoflurane upregulated CypD and then downregulated DCX, thus regulating migration, but how does CypD affect DCX is not clear, the authors should discuss this issue.

2.    The authors claim that CypD affects the mitochondrial permeability transit pore, I suggest that the authors should detect the effect of Sevoflurane on the mitochondrial permeability transit pore in NPC.

3.    In this paper, the authors should explain why they check ROS, what is the relation between ROS and cell migration? How dose CypD affects ROS?   

4.    The picture in Fig3B is not clear, I suggest the authors to revise this figure.

5.    The cell types should be labeled in detail in Fig4A, 4B and 4C.

6.    In the title of the article, I suggest that the authors should use the full name of the NPC rather than the abbreviation.

Author Response

Review 1 :

Comments: In this article, the authors found that sevoflurane inhibited NPC migration via CypD, which is interesting, but this paper also suffers from the following shortcomings.

 1. In the article, the authors found that Sevoflurane upregulated CypD and then downregulated DCX, thus regulating migration, but how does CypD affect DCX is not clear, the authors should discuss this issue.

Response: We thank the reviewer for the excellent advice. We have discussed the issue in the revised manuscript. The following paragraph has been included in the revised manuscript.

“It is not known at present how does CypD affect DCX levels. We hypothesize that the increases in CypD levels can cause mitochondrial dysfunction, which attenuates the migration of NPCs, marked by decreases in DCX levels. The future studies will test this hypothesis both in NPCs in vitro and in young mice in vivo.” [Discussion, page 10, the 6th paragraph].

2. The authors claim that CypD affects the mitochondrial permeability transit pore, I suggest that the authors should detect the effect of Sevoflurane on the mitochondrial permeability transit pore in NPC.

Response: We apologize for not being able to clearly state the published work in the original submission of the manuscript. Our previous studies have shown that sevoflurane can open mPTP in NPC. The following sentence has been included in the Introduction of the revised manuscript.

“Finally, in our previous work, we have detected the effect of sevoflurane on the mitochondrial permeability transit pore (mPTP) in NPCs, and demonstrated that sevoflurane reduced the binding of CypD with ANT (the other component of mPTP) and caused the mPTP opening [14]”. [Introduction, page 2, the 5th paragraph].

 3.In this paper, the authors should explain why they check ROS, what is the relation between ROS and cell migration? How dose CypD affects ROS? 

Response: ROS are critical for maintaining cellular homeostasis and function when produced in physiological ranges. Previous studies have suggested that CypD can regulate ROS generation. The following paragraph has been included in the revised manuscript.

“Previous studies have shown that mitochondrial ROS generation is dependent on CypD expression [34]. Thus, in the present study, we determined the effects of sevoflurane on levels of both ROS and CypD in NPCs.” [Introduction, page 2, the 7th paragraph].

  1. The picture in Fig3B is not clear, I suggest the authors to revise this figure.

Response: We have revised Figure 3B in the revised manuscript.

  1. The cell types should be labeled in detail in Fig 4A, 4B and 4C.

Response: We have labeled the cell types in Figure 4 of the revised manuscript.

  1. In the title of the article, I suggest that the authors should use the full name of the NPC rather than the abbreviation. 

Response: We have used the full name of the NPC in the revised title of manuscript.

Reviewer 2 Report

The manuscript by Lu et al. aimed to study the impact of sevoflurane anesthesia exposure on neural progenitor cell (NPC) migration. Their results suggest cyclophilin D (CypD) and doublecortin (DCX) as important players in anesthesia-induced risk of neurotoxicity in the developing brain. The impact of anesthetic agents on a developing brain is incompletely understood and limited in a number of studies. Therefore, the manuscript is an important addition to this topic and provides a better understanding of the mechanisms of developing brain cell response to anesthesia exposure.  

Overall, the work is interesting and well set. The following points might be considered to improve the manuscript before publication.  

1.     Could the authors further elaborate the importance of cell migration for cognitive function and brain development?

2.     Do CypD expression levels affect cell proliferation? Could the authors discuss this?

3.     Figure 4B shows increased levels of DCX in sevoflurane treated CypD knockout NPC cells when compared to Control. Could the authors address this result?

4.     The description under the Figure 4A-E legend is misinforming. Please, carefully revise and correct it.

5.     The description under the Figure 5E-G legend also needs to be revised and corrected.

6.     Adding all proteins with FDR<1 found in mitochondrial proteome analysis, in the supplementary table, will provide a great source of data for other researchers.

7.     Figure 5B would benefit from a bigger font since it is difficult to read. Also, the x-axis in Figure 5 needs to be labeled.

8.     Table 1 has cut words in columns and needs to be properly formatted.

Author Response

Comments: The manuscript by Lu et al. aimed to study the impact of sevoflurane anesthesia exposure on neural progenitor cell (NPC) migration. Their results suggest cyclophilin D (CypD) and doublecortin (DCX) as important players in anesthesia-induced risk of neurotoxicity in the developing brain. The impact of anesthetic agents on a developing brain is incompletely understood and limited in a number of studies. Therefore, the manuscript is an important addition to this topic and provides a better understanding of the mechanisms of developing brain cell response to anesthesia exposure. Overall, the work is interesting and well set. The following points might be considered to improve the manuscript before publication. 

  1. Could the authors further elaborate the importance of cell migration for cognitive function and brain development?

Response: We thank the reviewer for the good comments and excellent advice. We have discussed the issue in the revised version.

“NPCs migration is the prerequisite for the development of embryonic nervous system. In both embryonic and adult nervous systems, most NPCs need to migrate a certain distance to reach their functional sites [36]. Defective NPCs migration is closely associated with multiple neurodegenerative diseases and cognitive impairments [37, 38].” [Discussion, page 10, the 7th paragraph].

  1. Do CypD expression levels affect cell proliferation? Could the authors discuss this?

Response: We thank for the good suggestion and we have included the following paragraph in the revised manuscript.

In our previous study, we demonstrated that the sevoflurane anesthesia increased CypD levels and reduced NPCs proliferation in WT young mice or NPCs harvested from WT mice, but not in CypD KO mice or NPCs harvested from CypD KO mice. These data suggest that the sevoflurane might induce a CypD-dependent inhibition of cell proliferation [14]. In the present study, we further found that sevoflurane might also induce a CypD-dependent inhibition of migration of NPCs. [Discussion, page 14, the 8th paragraph].

  1. Figure 4B shows increased levels of DCX in sevoflurane treated CypD knockout NPC cells when compared to Control. Could the authors address this result?

Response: The quantification of Western blot showed that sevoflurane did not increase DCX levels in the CypD knockout NPC cells. We have revised the western blot image in the revised manuscript.

  1. The description under the Figure 4A-E legend is misinforming. Please, carefully revise and correct it.

Response: We apologize for the misinforming and have corrected the errors in the revised manuscript.

  1. The description under the Figure 5E-G legend also needs to be revised and corrected.

Response: We apologize and have corrected the errors of Figure 5 legend in the revised manuscript.

  1. Adding all proteins with FDR<1 found in mitochondrial proteome analysis, in the supplementary table, will provide a great source of data for other researchers.

Response: We thank the reviewer for the excellent advice. We have added the data in the Supplementary Table 1 of the revised manuscript.

  1. Figure 5B would benefit from a bigger font since it is difficult to read. Also, the x-axis in Figure 5 needs to be labeled. 

Response: We thank the reviewer for the good advice and have modified the figure 5B with bigger font in the revised manuscript and added the x-axis in figure 5B.

  1. Table 1 has cut words in columns and needs to be properly formatted. 

Response: We apologize and have reformatted the columns in the revised manuscript.

Reviewer 3 Report

The aim of this research study is to evaluate the sevoflurane exposure on neural progenitor cell (NPC) and wild and CypD (cyclophilin D) knockout mice. The authors of this study analysed the potentail changes in mitochondrial proteins and NPC migration by in vitro, molecular techniques and quantitative proteomic analysis.

This research study is interesting because several studies have now associated anesthetic exposure with alterations at the level of the nervous system, potentially leading to neurotoxicity and especially affect in neurodevelopment. In addition, several research studies also describe the role of oxidative stress in the generation of neurodegenerative diseases.

Overall, the manuscript clearly describes the scientific findings with related techniques. Although with some deficiencies in the structure and experimental design.

Major comments

1) The work you have done is comprehensive with a lot of methodology, but the abstract should be modified. Authors must follow the rules laid down by the journal the International Journal of Molecular Sciences journal's guidelines. It is recommended to use the word file "IJMS Microsoft Word template file", which you can download from the following page: https://www.mdpi.com/journal/ijms/instructions.

- Keywords must be added.

- Lu et al, not described cognitive assessment on the animals. Their conclusions must be based on the results and techniques they used.

2) Overall, the manuscript clearly describes the scientific findings with related experimental data. Although I find serious shortcomings in some aspects such as:

- The authors have not followed a proper order of presentation of the manuscript. It is difficult to understand and follow. Many paragraphs are mixed up e.g. acknowledgements, authors' contribution are after discussion when they should be at the end.

- Authors should modify the text to make it understandable. It is difficult to follow as they do not exactly detail the experimental design, i.e. it is not very clear what the authors want to convey. It seems to me that they have carried out both in vivo and in vitro studies.

3) In the Introduction section. Lu et al. should rewrite this section, as many citations are not what the authors cite. They cite references where they indicate in vivo studies when they also refer to in vitro studies. In addition, they also reference studies performed in human cells and only indicate in the text that they are studies performed in rodents.

4) In the materials and methods section. This section should also be modified. Regarding the experimental design and the choice of doses they used, I have questions:

- Lu et al. The authors should describe this section better, as it is not fully understood when conducting the experiments. They should describe in sections firstly for the in vitro experiments with NPCs and secondly the methods performed for the in vivo experiments.

- I have observations and questions about the in vitro study. The authors should describe the treatment used with the NPCs. It is not clear whether they first performed the sevoflurane treatment on the animals or whether they cultured NPCs from the animals (mice).

In any case, on page 12, line 354, the authors indicate that the PCNs were treated with 4.1% sevoflurane for 4 hours and cite other work that was done with isoflurane. I would have liked to read in more detail how these cells were treated and that a dose response was performed to choose the dose to be used in subsequent experiments. For example, Yang et al., 2017 and Zhao et al., 2013 adequately describe the exposure of anesthetic to cells.

Here are the references that might be useful to you in correcting this paragraph. Also taking into consideration that isoflurane and sevoflurane have different effects on cell cultures (Wang et al., 2008).

Wang QJ. et al., 2008. Different effects of isoflurane and sevoflurane on cytotoxicity. Chinese Medical Journal 121(4), 341-346. https://journals.lww.com/cmj/Fulltext/2008/02020/Different_effects_of_isoflurane_and_sevoflurane_on.12.aspx

Zhou YF. et al., 2016. Autophagy activation prevents sevoflurane-induced neurotoxicity in H4 human neuroglioma cells. Acta Pharmacologica Sinica 37, 580-588. https://doi.org/10.1038/aps.2016.6

Piao et al., 2020. Sevoflurane Exposure Induces Neuronal Cell Parthanatos Initiated by DNA Damage in the Developing Brain via an Increase of Intracellular Reactive Oxygen Species. Frontiers in Cellular Neuroscience 14, 583782. https://doi.org/10.3389/fncel.2020.583782

Yang et al., 2017. Sevoflurane decreases self-renewal capacity and causes c-Jun N-terminal kinase–mediated damage of rat fetal neural stem cells. Scientific Reports 7, 46304. https://doi.org/10.1038/srep46304

Song et al., 2017. Maternal Sevoflurane Exposure Causes Abnormal Development of Fetal Prefrontal Cortex and Induces Cognitive Dysfunction in Offspring. Stem Cells International 2017: 6158468. https://doi.org/10.1155/2017/6158468

Zhao et al., 2013. Dual Effects of Isoflurane on Proliferation, Differentiation, and Survival in Human Neuroprogenitor Cells. Anesthesiology 118, 537-549. https://doi.org/10.1097/ALN.0b013e3182833fae

5) In the section Results and Discussion. It is important to mention that the Discussion should be improved according to results and using current references and with a relevant quality index.

Minor comments

- I would also suggest that the author revised the grammar of the manuscript to increase the English level of the manuscript.

- Please cite references in the text according to the guidelines of the International Journal of Molecular Sciences. It is also necessary to review the bibliography, as it does not correspond to the format of this journal.

- I suggest you check the abbreviations throughout the manuscript.

Recommendation

For the reasons stated above, I recommend that this work be Reconsider after major revision and  I hope the outcome of this specific submission will not discourage you from the submission of future manuscripts.

I also attach some comments in the PDF file

Author Response

Comments: The aim of this research study is to evaluate the sevoflurane exposure on neural progenitor cell (NPC) and wild and CypD (cyclophilin D) knockout mice. The authors of this study analyzed the potential changes in mitochondrial proteins and NPC migration by in vitro, molecular techniques and quantitative proteomic analysis. This research study is interesting because several studies have now associated anesthetic exposure with alterations at the level of the nervous system, potentially leading to neurotoxicity and especially affect in neurodevelopment. In addition, several research studies also describe the role of oxidative stress in the generation of neurodegenerative diseases. Overall, the manuscript clearly describes the scientific findings with related techniques. Although with some deficiencies in the structure and experimental design.

Major comments

  1. The work you have done is comprehensive with a lot of methodology, but the abstract should be modified. Authors must follow the rules laid down by the journal the International Journal of Molecular Sciences journal's guidelines. It is recommended to use the word file "IJMS Microsoft Word template file", which you can download from the following page: https://www.mdpi.com/journal/ijms/instructions.

Response: We apologize and have modified the manuscript according to the IJMS guidelines in the revised manuscript.

  1. Keywords must be added.

Response: Keywords have been added in the revised manuscript.

  1. Lu et al, not described cognitive assessment on the animals. Their conclusions must be based on the results and techniques they used.

Response: We thank for the good comment and the conclusions have been revised as the following:

“In conclusion, our studies suggest that anesthetic sevoflurane can inhibit migration of NPCs and such effects are dependent on CypD. These results could promote further research into mitochondria, migration of NPCs, and anesthesia-induced neurotoxicity in the young brain, which could ultimately lead to better postoperative outcomes in children.” [Discussion, page 11, the 2nd paragraph].

  1. Overall, the manuscript clearly describes the scientific findings with related experimental data. Although I find serious shortcomings in some respects such as:

- The authors have not followed a proper order of presentation of the manuscript. It is difficult to understand and follow. Many paragraphs are mixed up e.g., acknowledgements, authors' contribution are after discussion when they should be at the end. 

Response: We apologize and have significantly revised manuscript.

 Authors should modify the text to make it understandable. It is difficult to follow as they do not exactly detail the experimental design, i.e., it is not very clear what the authors want to convey. It seems to me that they have carried out both in vivo and in vitro studies. 

Response: The studies included both in vitro and in vivo studies. In the revised manuscript, we have stated that we firstly did the in vitro experiments with cultured NPCs and then carried out in vivo relevance studies in young mice. The following sentences have been modified and included in the revised manuscript.

“For the in vitro studies, the NPCs were treated with 4.1% sevoflurane plus 21% O2 and 5% CO2 for 4 hours, as described in previous studies [14].” [Methods, page 11, the 5th paragraph].

“The mice in the anesthesia group received 3% sevoflurane plus 60% oxygen (balanced with nitrogen) for two hours at postnatal day (P)6, P7, and P8, as described in our previous studies [39-41, 43].” [Methods, page 13, the 1st paragraph].

  1. In the Introduction section. Lu et al. should rewrite this section, as many citations are not what the authors cite. They cite references where they indicate in vivo studies when they also refer to in vitro studies. In addition, they also reference studies performed in human cells and only indicate in the text that they are studies performed in rodents.

Response: We truly thank the Reviewer to point out the errors of the manuscript. We have re-written the majority of the Introduction section with correct references in the revised manuscript.

  1. In the materials and methods section. This section should also be modified. Regarding the experimental design and the choice of doses they used, I have questions: 

- Lu et al. The authors should describe this section better, as it is not fully understood when conducting the experiments. They should describe in sections firstly for the in vitro experiments with NPCs and secondly the methods performed for the in vivo experiments.

Response: We have modified the materials and methods section in the revised manuscript.

  1. I have observations and questions about the in vitro study. The authors should describe the treatment used with the NPCs. It is not clear whether they first performed the sevoflurane treatment on the animals or whether they cultured NPCs from the animals (mice). 

Response: We first cultured NPCs from the mice, and then treated the NPCs with sevoflurane anesthesia. We have significantly revised the manuscript. The suggested references have been included in the revised manuscript.

  1. In any case, on page 12, line 354, the authors indicate that the NPCs were treated with 4.1% sevoflurane for 4 hours and cite other work that was done with isoflurane. I would have liked to read in more detail how these cells were treated and that a dose response was performed to choose the dose to be used in subsequent experiments. For example, Yang et al., 2017 and Zhao et al., 2013 adequately describe the exposure of anesthetic to cells.

Here are the references that might be useful to you in correcting this paragraph. Also taking into consideration that isoflurane and sevoflurane have different effects on cell cultures (Wang et al., 2008).

Wang QJ. et al., 2008. Different effects of isoflurane and sevoflurane on cytotoxicity. Chinese Medical Journal 121(4), 341-346. https://journals.lww.com/cmj/Fulltext/2008/02020/Different_effects_of_isoflurane_and_sevoflurane_on.12.aspx

Zhou YF. et al., 2016. Autophagy activation prevents sevoflurane-induced neurotoxicity in H4 human neuroglioma cells. ActaPharmacologicaSinica 37, 580-588. https://doi.org/10.1038/aps.2016.6

Piao et al., 2020.Sevoflurane Exposure Induces Neuronal Cell Parthanatos Initiated by DNA Damage in the Developing Brain via an Increase of Intracellular Reactive Oxygen Species. Frontiers in Cellular Neuroscience 14, 583782. https://doi.org/10.3389/fncel.2020.583782

Yang et al., 2017.Sevoflurane decreases self-renewal capacity and causes c-Jun N-terminal kinase–mediated damage of rat fetal neural stem cells. Scientific Reports 7, 46304. https://doi.org/10.1038/srep46304

Song et al., 2017. Maternal Sevoflurane Exposure Causes Abnormal Development of Fetal Prefrontal Cortex and Induces Cognitive Dysfunction in Offspring. Stem Cells International 2017: 6158468. https://doi.org/10.1155/2017/6158468

Zhao et al., 2013.Dual Effects of Isoflurane on Proliferation, Differentiation, and Survival in Human Neuroprogenitor Cells. Anesthesiology 118, 537-549. https://doi.org/10.1097/ALN.0b013e3182833fa 

Response: We thank the reviewer for the great comments and excellent advice. In our previous work, we have proved that 4.1% sevoflurane for 4 hours could increase CypD levels and cause mitochondrial dysfunction. Therefore, we choose this dose of sevoflurane to be used in subsequent experiments. We have corrected the citation in the revised manuscript.

  1. In the section Results and Discussion. It is important to mention that the Discussion should be improved according to results and using current references and with a relevant quality index.

Response: We have significantly revised the Discussion section.

Minor comments

- I would also suggest that the author revised the grammar of the manuscript to increase the English level of the manuscript.

 Response: We have improved the grammar in the revised manuscript.

 - Please cite references in the text according to the guidelines of the International Journal of Molecular Sciences. It is also necessary to review the bibliography, as it does not correspond to the format of this journal.

Response: We have revised the references in the revised manuscript

- I suggest you check the abbreviations throughout the manuscript.

Response: We thank the reviewer for the good advice and have checked and corrected the abbreviations in the revised manuscript.

Round 2

Reviewer 1 Report

The manuscript has been sufficiently improved to warrant publication in IJMS.

Author Response

Thank you very much for your positive feedback regarding the revised manuscript. We appreciate your advices and valuable input and guidance throughout the review process, which has been instrumental in enhancing the quality of our work.

Reviewer 3 Report

Dear authors,

Thank you very much for your replies. The manuscript has been improved according to my observations. However, there are minor details that need to be corrected and that I mentioned to you in initial version 1 of the manuscript:

- Page 12, line 347: Please change "1 million" for "1 x 106"

- Page 12, line 348: Please change "plated" for "seeded"

-Page 12, line 353: Please change "plated" for "seeded": Please change this sentence to: "wound healing and transwell assays"

- Point 4.3. Wound healing assay, Page 12 line 373: Please describe in more detail this method: Number of cells used, Did they use serum starvation and a substance that blocks cell proliferation?, Counted cells at the beginning and end of the experiment?

- Page 12, line 375: "in vitro", This word is repetitive wound healing assay is a standard in vitro technique for probing collective cell migration

- Page 12, lines 379-381: Please indicate how this measurement was made. Did you use any software? 

- Page 12, line 382:  Please consider changing to the following sentence: "Transwell migration assay"

- Page 12, line 382:  Please consider changing to the following sentence: "Transwell migration assay"

- Page 12, line 386:  Please change "1 x 106" for "1 x 106"

- Page 13, line 395:  4.5. Western blot analysis: Please indicate the number of experiments performed fo example "Each band in the Western blot represented an independent experiment. The results were averaged from three to 8 independent experiments."

- Page 13, line 399:  Please change "western" for "Western"

- Page 13, lines 400-401:  Please delete the sentence "We present changes in protein amounts in the NPCs as a percentage of those in control"

- Page 13, lines 407-414:  Please modify this paragraph, detailing the technique used. 

Modify the sentence "were placed in a clear 96-well cell culture plate overnight in the incubator", because Black plates are normally used to measure ROS by fluorescence

What is the final concentration of DCFH-DA you used?

In the sentence "Finally, the fluorescence was read 418 with a fluorometric plate reader at 480 nm/530 nm." At what time(s) did they measure ROS production after adding the probe?

- Page 13, line 428:  Please modify "(P)6" for "P6". 

- Page 13, line 430:  Please modify "Harvest of brain tissues" for "Brain tissue collection". 

- Page 14, line 478:  Please modify "0.05 (*) and 0.01 (**)," for " P < 0.05 and P < 0.01". 

Author Response

-Page 12, line 347: Please change "1 million" for "1 x 106"

Response: We thank the reviewer for the detail advice. We have changed "1 million" for "1 x 106" in current revised version in Page 11, line 353.

- Page 12, line 348: Please change "plated" for "seeded"

Response: We thank the reviewer for the detail advice. We have changed "plated" for "seeded" in current revised version in Page 11, line 354.

-Page 12, line 353: Please change "plated" for "seeded": Please change this sentence to: "wound healing and transwell assays"

Response: We thank the reviewer for this advice. We have changed this sentence to “For the in vitro studies” to include wound healing, transwell assays, ROS, and WB, in current revised version in Page 11, line 359.

- Point 4.3. Wound healing assay, Page 12 line 373: Please describe in more detail this method: Number of cells used, Did they use serum starvation and a substance that blocks cell proliferation.

Response: We thank the reviewer for this advice. We have modified this paragraph and added detail information regarding this part as following “The NPCs were seeded on poly-D-lysine (PDL) precoated six-well plates at a cell density of 1.0×106 cells/ml in "complete" proliferation medium and allowed to attach for 12 hours. After cell attachment, plates with consistent cell density were selected for the next experiments. After cell attachment, a scratch was created on the monolayer using a sterile plastic 200μl micropipette tip in each cultured well. The cells were washed twice in warm serum-free medium to remove cellular debris and floating cells from scratches, and then exposed to 4.1% sevoflurane plus 21% O2 and 5% CO2 for 4 hours. The scratch wounds were visualized with a phase contrast light microscope. Photographs were captured at two time points (0 hours and 24 hours after sevoflurane exposure). The NPCs were kept in the incubator between the photographs with in "complete" proliferation medium. The measurement was conducted by using Image J software. The distance between the two separated sides at 0 hours was used as a reference, and subsequent changes in the distance over time were measured as the cell migration distance. Result was quantified by determining the percent scratch closed [ (initial scratch width- final scratch width) /initial scratch width x 100%].”

In our previous experiment, after cell attachment, we tried to change the "complete" proliferation medium to NeuroCult™ neural stem cell basal media (serum free) to inhibit cell proliferation. It may due to the both the treatment and long experiment duration, we lost >50% cells at the end of experiment timepoint. So, in current studies, we used "complete" proliferation medium.

- Page 12, line 375: "in vitro", This word is repetitive wound healing assay is a standard in vitro technique for probing collective cell migration. 

Response: We deleted “in vitro” in revised version, Thank you.

- Page 12, lines 379-381: Please indicate how this measurement was made. Did you use any software?

Response: We thank the reviewer for this advice. We added the following sentence in revised version “The measurement was conducted by using Image J software.”

- Page 12, line 382:  Please consider changing to the following sentence: "Transwell migration assay."

Response: We thank the reviewer for this advice. We have revised this.

- Page 12, line 386:  Please change "1 x 106" for "1 x 106"

Response: We thank the reviewer for this advice. We have revised this.

- Page 13, line 395:  4.5. Western blot analysis: Please indicate the number of experiments performed for example "Each band in the Western blot represented an independent experiment. The results were averaged by six independent experiments."

Response: We thank the reviewer for this advice. We added the following sentence in revised version “Each band in the Western blot represented an independent experiment. The results were averaged by six independent experiments.”

- Page 13, line 399:  Please change "western" for "Western".

Response: We thank the reviewer for this advice. We have revised this.

- Page 13, lines 400-401:  Please delete the sentence "We present changes in protein amounts in the NPCs as a percentage of those in control."

Response: We thank the reviewer for this advice. We have revised this.

- Page 13, lines 407-414:  Please modify this paragraph, detailing the technique used.

Response: We thank the reviewer for this advice. We have modified this paragraph and added detail information regarding this part as following “Briefly, prepare NPCs cells by culturing them in a black 96-well cell culture plate overnight in the incubator. Prepare the DCFH-DA/media solution by dissolving DCFH-DA in the cell culture media to a final concentration of 10 μM. Protect the solution from light during the preparation and use. Remove the cell culture media from the NPCs cells and add the DCFH-DA/media solution to each well. Incubate the cells at 37°C for 30 minutes to allow the cells to uptake the DCFH-DA. After 30 minutes, remove the DCFH-DA/media solution and wash the cells with fresh media to remove any excess DCFH-DA. The DCFH-DA loaded NPCs cells were then exposed to 4.1% sevoflurane for 4h. The treated cells were lysed by adding 100 μl of cell lysis buffer to each well. Mix the lysate thoroughly and incubate for 5 minutes at room temperature away from light. Finally, the fluorescence was read with a fluorometric plate reader at 480 nm/530 nm within 10-15min. The amount of fluorescence is proportional to the amount of ROS present in the cells.”

- Page 13, line 428:  Please modify "(P)6" for "P6".

Response: We thank the reviewer for this advice. We have revised this.

- Page 13, line 430:  Please modify "Harvest of brain tissues" for "Brain tissue collection".

Response: We thank the reviewer for this advice. We have revised this.

- Page 14, line 478:  Please modify "0.05 (*) and 0.01 (**)," for " P < 0.05 and P < 0.01".

Response: We thank the reviewer for this advice. We have revised this.
